# Binning of Linear Models for Time Series Forecasting

## Abstract

Linear models have been gaining attention in the time series forecasting (TSF) field, as several variations of modified linear layers have shown excellent performance across benchmarked datasets. This work extends these efforts by exploring the application of the binning technique to linear models. The conceptual rationale is based on the idea that binning can serve as a simple means of efficient learning through isolating temporal patterns. Instead of relying on temporally adjacent observations, binning adopts another perspective by exposing binned linear layers to periodically grouped data, treating temporal neighbors independently. Both conceptual analysis and empirical experimentation are conducted, with the results demonstrating that this modification can lead to improvements in prediction error. This work positions binning as a simple and effective technique, contributing to the exploration of representations structured by periodicity.

## 1 Introduction

Time series forecasting (TSF) has become one of the significant domains of applications of machine learning and deep learning. It plays an important role in daily operations with direct implementations in energy, financial, and other sectors (Benidis et al., 2022). A wide variety of research has been conducted in developing different types of models. Some of the recent advances and modifications include configuring transformer models to TSF settings (Zhou et al., 2021; 2022; Nie et al., 2022; Liu et al., 2023), constructing lightweight models with low parameter count (Si et al., 2025; Wang et al., 2025; Ekambaram et al., 2023), applying mixing (Wang et al., 2024c; Ekambaram et al., 2024; Wang et al., 2024b), and exploration of the channel strategies (Qiu et al., 2025b;a; Si et al., 2025).

One of the key distinctions is the properties of the time series data, which include the trend and the seasonal components. Configuring a model to attend to and learn the temporal dynamics is the essence of the TSF setting. To capture periodic properties, some of the proposed ideas include learning in the frequency domain (Wang et al., 2024a), transforming into a 2D space based on multiple periods (Wu et al., 2022), enhancing convolution layers with variable-independent embeddings (Luo & Wang, 2024), and a variety of other techniques.

This work explores a different way to interpret seasonal and trend data in the TSF setting. The technique is inspired by a regressive setting with exogenous variables in the energy sector. One of the common modeling techniques involves binning individual regressors according to the data periodicity (Hu et al., 2023). This step isolates all temporal dependencies to regress endogenous observations exclusively on the exogenous variables at the time step of the period. This is typically done for the instance of load modeling, in which the observed temperature can explain the majority of the endogenous variable variation. Inspired by the efficiency of the technique, this work reconfigures binning to the typical TSF setting with a lookback window and prediction horizon.

Recently, linear models have been receiving attention in the TSF setting. Several works found that with simple preprocessing techniques, linear models are capable of achieving performance on par with complex architectures such as transformers (Zeng et al., 2023; Toner & Darlow, 2024). The particular attractiveness lies in their simplicity, computational efficiency, and more transparent interpretability. Some of the recent works combine linear layers with modifications, such as mixing, recurrent cycle modeling, and frequency domain interpolation, to further enhance the performance of linear models in the TSF setting Lin et al. (2024a); Xu et al. (2023).

To test the proposed binning modification, linear models are used as they previously showed high performance. Both conceptual and empirical experiments are conducted, with the results showing improvements across the majority of the benchmarked datasets.

Hence, the contributions of this work are the following:

- The binning technique is introduced for the TSF setting with a conceptual rationale.
- The technique is applied to linear models. The proposed models are tested on mainstream literature datasets for appropriate benchmarking and empirical experimentation.
- The results are discussed and analyzed with the important implications outlined.

This work can serve as an initial step towards the exploration of binning in the TSF setting. The results of this study are intended to inspire further research into theoretical and experimental advances of the proposed technique.

## 2 RELATED WORK

Transformers have widely been configured to the TSF setting (Deng et al., 2024; Zhang & Yan, 2023; Woo et al., 2022). One of the similar ideas to binning is the attention mechanism in the Autoformer model (Wu et al., 2021). In the work, the authors propose to calculate attention scores based on the autocorrelation function and Fourier transforms, which would capture the seasonal and trend behavior of the data. The model attends to the similar points according to the periodicity.

Among light-weight models, the TiDE model (Das et al., 2023) leverages encoder-decoder mapping of lookback window and covariates to capture temporal relationships; the LightTS model (Zhang et al., 2022) applies interval and continuous sampling with an MLP backbone; The SparseTSF model (Lin et al., 2024b) leverages periodic sampling strategies to capture temporal dependencies. While both SparseTSF and binning utilize periodicity for grouping data, they differ in several aspects. SparseTSF applies a sliding convolution before sampling, ensuring that each sampled chunk incorporates information from neighboring time points. Additionally, SparseTSF processes entire chunks through linear layers, whereas binning processes individual periodic points. One form of binning, denoted as discretization, is proposed in (Rabanser et al., 2020). The approach proposed primarily uses value-based quantile binning as a normalization technique for deep learning models. This work introduces period-aligned binning for linear models, where bins correspond to specific phases within temporal cycles rather than value ranges.

SparseTSF demonstrates that periodic decomposition can be highly effective when combined with temporal smoothing and chunk-level processing. This raises a question: to what extent can periodicity-based approaches succeed with different design choices? Binning explores a complementary direction by examining whether focusing purely on periodic relationships — without temporal aggregation — can yield satisfactory predictions. This approach treats each period independently and poses the question: *Can effective forecasting be achieved by training separate linear layers on periodically grouped data, rather than processing the entire lookback window sequence?* While both SparseTSF and binning leverage periodicity, they represent different points on the spectrum of how much temporal context to preserve versus how much to rely on periodic structure alone. This work seeks to investigate the relative importance of context preservation and periodic reliance through conceptual and empirical analysis.

## 3 METHODOLOGY

### 3.1 PRELIMINARIES

The notation used in Lin et al. (2024b) is followed in this work. Let $L$ denote a lookback window, $H$ denote the forecast horizon, and $C$ denote the number of channels. The TSF setting is formulated as $\bar{x}_{t+1:t+H} = f(x_{t-L+1:t})$, where $x_{t-L+1:t} \in \mathbb{R}^{L \times C}$ and $\bar{x}_{t+1:t+H} \in \mathbb{R}^{H \times C}$. This says that, given a lookback window, the model makes predictions over a selected horizon period. The period is denoted as $w$ and satisfies the condition of $s(t) = s(t + w)$, where $s(t)$ denotes a periodic component of the time series.

### 3.2 BINNING

**Definition 1.** *Let $w$ be the period length, and let $S_b \subseteq \{0, 1, 2, \ldots, w - 1\}$ denote the index set associated with bin $b$. Given a time $t$ and lookback window length $L$, the sampled points for bin $b$ from the lookback window $x_{t-L+1:t}$ are defined as:*

$$x_{t-L+1:t}^b = \left\{ x_{(k-1)w+i} \mid i \in S_b,\ k \in \mathbb{Z},\ t - L + 1 \leq (k-1)w + i \leq t \right\}.$$

*where $k$ is the period index and $i$ is an offset within the period.*

In this work, the assumption is that $w$ period equals the number of bins $b$. Hence, $w/b = 1$, meaning that for every period cycle, each bin samples only one point [1]. This definition can be used to reformulate how forecasting is done through binning.

**Lemma 1.** *For each bin, the forecasting setting can be reformulated as:*

$$\bar{x}_{t+1:t+H}^b = f(x_{t-L+1:t}^b) \tag{1}$$

*where binned sampling and prediction are done according to the Definition (1).*

The proposed technique assumes that the bins process respective periodic data and make predictions independently. The input is processed for each bin before passing to the respective function, and the predictions are stacked (concatenated) at the end to obtain the full horizon prediction vector. The pseudocode for how forecasting is conducted can be found in 1. Another assumption is that each bin has an identical function $f(\cdot)$. This definition can be easily extended to the case of individual bin functions $f^b(\cdot)$.

---

**Algorithm 1** Binned Forecasting Pseudocode

    Number of bins $b$, period $w$, function for each bin $f(\cdot)^b$, lookback $L$, horizon $H$
1: Observe full lookback sequence $x_{t-L+1:t}$
2: **for** $i = 1$ to $b$ **do**
3:     Obtain input for the bin $i$ as $x_{t-L+1:t}^b$
4:     Pass the input through the function $f(\cdot)$ of the respective bin
5:     Obtain prediction $\bar{x}_{t+1:t+H}^b$
6:     Stack (concatenate) prediction to $\bar{x}_{t+1:t+H}$
7: **end for**
8: Obtain a full prediction horizon vector $\bar{x}_{t+1:t+H}$

---

### 3.3 RATIONALE

The majority of the literature represents a time series in the following way:

$$x(t) = s(t) + p(t) \tag{2}$$

where $s(t)$ is a periodic (seasonal) component and $p(t)$ is a trend component.

Using the definition of binning, the forecasting can be applied to this representation of the time series.

**Theorem 1.** *Let the time series be given by $x(t) = s(t) + p(t)$. Then, for each bin $b$, the binned time series $x^b(t) = x(kw + b) = s(kw + b) + p(kw + b)$, has the form $x^b(t) = c_b + p^b(t)$, where $c_b = s(b)$ is a constant (i.e., the value of the seasonal component at phase $b$) and $p^b(t) = p(kw + b)$ is a resampled version of the trend component.*

The seasonal component is eliminated within each bin, and each binned time series follows the trend with a bin-specific intercept shift. This implies that binning the time series can serve as another way

---

[1]This assumption resonates with some of the discussion made in (Lin et al., 2024b). If the data are periodic, periodicity can potentially be identified through the ACF function, and for certain domains, the periodicity might be known. For other cases and practical purposes, $w$ can be treated as a hyperparameter.

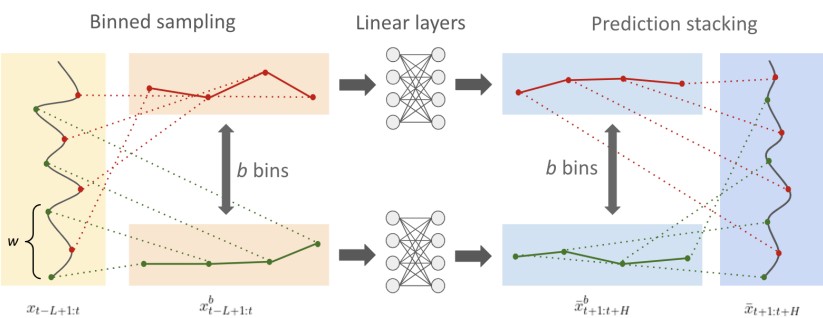

Figure 1: Binning architecture

of observing the seasonal and trend components. The periodic relationship is treated independently of the temporally adjacent points. Each bin is only modeled to predict the trend using the respective function.

This representation can be extended. Suppose the representation of the time series is in the following way:

$$x(t) = s(t)p_1(t) + p_2(t) \tag{3}$$

The periodic component is complicated by the individual trend dynamics $p_1(t)$. The additional trend $p_2(t)$ is independent of the periodic trend.

**Theorem 2.** *Let the time series be given by:* $x(t) = s(t) \cdot p_1(t) + p_2(t)$*, where $s(t)$ is a periodic function with period $w$ (i.e., $s(t + w) = s(t)$), and $p_1(t)$ and $p_2(t)$ are trend components. Then, for each bin $b$, the binned time series has the form:* $x^b(t) = c_b \cdot p_1^b(t) + p_2^b(t)$*, where $c_b = s(b)$ is a constant, $p_1^b(t) = p_1(kw + b)$ and $p_2^b(t) = p_2(kw + b)$ are resampled versions of the trend components.*

Binning removes the seasonal variability within each bin, and the remaining signal reflects phase-dependent trend behavior. Hence, the conceptual rationale is that binning can help model and represent more complex time series, such as in Equation 3, through simple periodic sampling. The extension for a multiperiodic case is provided in the Appendix.

## 3.4 VALIDATION

The binning technique is not necessarily restricted to any function or architecture. Nevertheless, the high-level concept can be easily demonstrated and validated on the linear models. Additional motivation is that previous work showed that linear models struggle with capturing the trend relationship (Li et al., 2023). It was shown that Reversible Instance Normalization (RevIN) (Kim et al., 2021) can help eliminate challenges associated with learning the trend dynamics. For the time series, which exhibit behavior as in Equation 2, binning will not provide additional value as a simple linear model with RevIN can learn dynamics efficiently.

For the time series exhibiting the behavior of Equation 3, the temporal dynamics are more complex. The seasonal points exhibit an unequal trend, bringing potential challenges for the linear models to learn temporal dynamics. Hence, the above rationale becomes additionally valuable as binning can serve as a means for individual linear models to learn trends efficiently. Essentially, because each bin of data only contains its own trend, the associated linear function can capture that trend successfully. The visualization of the architecture is provided in Figure 1.

### 3.4.1 SYNTHETIC DATA VALIDATION

To complement the above rationale analysis, some theoretical experiments are conducted. Using Equation 3, the following definition is introduced.

**Definition 2.** *Suppose $s(t)$ is $K_s$-Lipschitz continuous, $p_1$ is $K_{p1}$-Lipschitz continuous, and $p_2$ is $K_{p2}$-Lipschitz continuous. Define the trend-to-oscillation ratio as $TO = \frac{K_s K_{p1}}{K_{p2}}$.*

This definition is used to compile experiments across functions with different dominating components. If $TO > 1$, the function is denoted as oscillation-dominated, and if $TO < 1$ the function is denoted as trend-dominated.

To conduct the experiments, the following function is used as the backbone of the synthetic dataset:

$$y = C_1 x + b + C_2 x \sin(\pi x / C_3) \qquad (4)$$

The constants are chosen such that different $TO$ ratios are tested. Four models are tested: a simple Linear model and RLinear, both without and with binning. Additional details on the setup are described in the Appendix.

### 3.4.2 EMPIRICAL BENCHMARKED DATASET VALIDATION

While binning can provide benefits to learn time series of the form of Equation 3, it could be argued that real-world datasets rarely exhibit such behavior, including lacking any periodicity. Hence, the empirical experiments are conducted on the mainstream datasets to test whether binning can provide any improvements. The choice of datasets is: ETTh1, ETTh2, Electricity (periodic) and ETTm1, ETTm2 (non-periodic).

The linear models, previously proposed in the literature, are tested in this work. In particular, the models of choice are Linear, NLinear, DLinear (Zeng et al., 2023), and RLinear (Li et al., 2023) . Intuitively, RLinear should have the most potential due to its efficient trend representation properties. Each bin can apply RevIN to the individual trend dynamics. Nevertheless, the other models are iterated to test whether binning provides improvements regardless of which linear model is picked. For instance, in case binned representations are more complex than just an individual trend, the DLinear model applies additional decomposition for efficient learning. Additionally, a binned Global RLinear model is introduced. This architecture applies RevIN to the entire input sequence before binning and the forward pass to an individual linear layer. Additional details on the setup are also provided in the Appendix.

## 4 RESULTS

### 4.1 SYNTHETIC DATA EXPERIMENTS

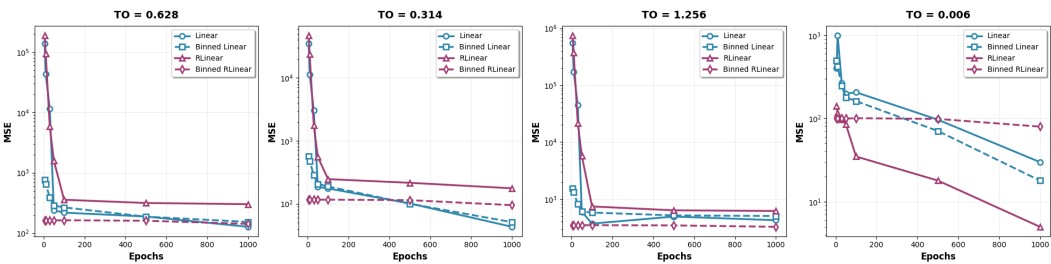

Figure 2: Synthetic data results

The results for synthetic data for different TO ratios are presented in Figure 3. The Binned RLinear model provides convergence even at a few epoch iterations, showcasing extreme efficiency in time series learning. The error stays constant for the model regardless of the number of epochs traversed. The Binned Linear also provides lower error compared to its unbinned counterpart for a low number of epochs. For the low TO (i.e., trend-dominated series), the RLinear model showcases great performance with longer training and performs better than all other models even after an insignificant number of iterations. This agrees with the theory developed in Li et al. (2023) that showed that

Table 1: Results on benchmarked datasets. The results are averaged over the prediction horizons $T \in \{96, 192, 336, 720\}$. The full results are provided in the Appendix. The **bold** indicates the best result and underline indicates the second best result in the respective column. The improvement indicates the improvement relative to the respective model without binning [for the percent improvement, the formula is (not_binned-binned)/(not_binned)*100%].

| Dataset | ETTh1 | | ETTh2 | | ETTm1 | | ETTm2 | | Electricity | |
|---|---|---|---|---|---|---|---|---|---|---|
| Metric | MSE | MAE | MSE | MAE | MSE | MAE | MSE | MAE | MSE | MAE |
| Linear | 0.431 | 0.444 | 0.438 | 0.449 | 0.370 | 0.394 | 0.302 | 0.362 | 0.167 | 0.269 |
| Linear+Bin | 0.420 | 0.430 | 0.493 | 0.484 | 0.366 | 0.387 | 0.282 | 0.345 | 0.166 | 0.261 |
| Improv. | 0.011 | 0.014 | -0.055 | -0.035 | 0.004 | 0.007 | 0.020 | 0.017 | 0.001 | 0.008 |
| Improv. (%) | 2.55 | 3.15 | -12.55 | -7.80 | 1.08 | 1.78 | 6.62 | 4.70 | 0.60 | 2.97 |
| NLinear | 0.414 | 0.424 | **0.347** | **0.391** | 0.375 | 0.392 | 0.264 | 0.321 | **0.162** | **0.256** |
| NLinear+Bin | 0.413 | **0.414** | 0.362 | 0.401 | 0.371 | 0.386 | **0.260** | **0.316** | 0.167 | 0.258 |
| Improv. | 0.001 | 0.010 | -0.015 | -0.010 | 0.004 | 0.006 | 0.004 | 0.005 | -0.005 | -0.002 |
| Improv. (%) | 0.24 | 2.36 | -4.32 | -2.56 | 1.07 | 1.53 | 1.52 | 1.56 | -3.09 | -0.78 |
| DLinear | 0.429 | 0.441 | 0.470 | 0.467 | 0.365 | 0.387 | 0.311 | 0.368 | 0.171 | 0.272 |
| DLinear+Bin | 0.417 | 0.428 | 0.489 | 0.480 | 0.373 | 0.392 | 0.291 | 0.350 | 0.166 | 0.259 |
| Improv. | 0.012 | 0.013 | -0.019 | -0.013 | -0.008 | -0.005 | 0.020 | 0.018 | 0.005 | 0.013 |
| Improv. (%) | 2.80 | 2.95 | -4.04 | -2.78 | -2.19 | -1.29 | 6.43 | 4.89 | 2.9 | 4.78 |
| RLinear | 0.417 | 0.427 | 0.359 | 0.400 | **0.364** | **0.381** | 0.263 | 0.320 | 0.170 | 0.266 |
| RLinear+Bin | 0.418 | 0.415 | 0.365 | 0.403 | **0.364** | **0.381** | 0.263 | 0.318 | 0.168 | 0.258 |
| RLinear+Bin+G | **0.411** | 0.419 | 0.363 | 0.405 | 0.370 | 0.386 | **0.260** | 0.317 | 0.167 | 0.258 |
| Improv. | 0.006 | 0.008 | -0.004 | -0.005 | 0 | 0 | 0.003 | 0.003 | 0.003 | 0.008 |
| Improv. (%) | 1.44 | 1.87 | -1.11 | -1.25 | 0 | 0 | 1.14 | 0.94 | 1.76 | 3.01 |

RLinear architecture can efficiently train on trend. Nevertheless, for other cases, RLinear actually does not perform better and performs worse than a regular Linear model, especially when trained for a higher number of epochs. This challenges the overall notion that RLinear provides significant improvements for a certain behavior of time series. While RLinear performs better than a regular Linear model for a low number of epochs, it does not perform better than any binned model unless the time series exhibits strong single-trend characteristics. Additionally, binning provides better predictions for a low number of training iterations, showing the efficiency of the proposed architecture.

## 4.2 EMPIRICAL BENCHMARKED DATASET EXPERIMENTS

The results for the benchmarked datasets are presented in Table 1. The binned models provide improvements compared to their unbinned counterparts for the majority of datasets except ETTh2, and a few exceptions for ETTm1 and Electricity datasets. Notable improvements are evident for the Linear and DLinear models. One of the results is that binning consistently provides worse predictions across the same dataset (ETTh2 in particular), regardless of what type of linear model is used. Notably, binning can also provide improvements for non-periodic datasets, such as ETTm1 and ETTm2.

## 5 DISCUSSION

### 5.1 REPRESENTATION LEARNING

One of the advantages of linear models is their simpler interpretability compared to more complex architectures. One of the possible interpretations is the exploration of model weights, which can be visualized to observe the lookback and sequence relationships. One of the examples is plotted in Figure 3.

It can be seen that since binning removes some of the edges between the nodes, the majority of the weights are zero. This representation can be interpreted as binning, removing some of the unnecessary training and noisy weights, which can be observed for the unbinned model.

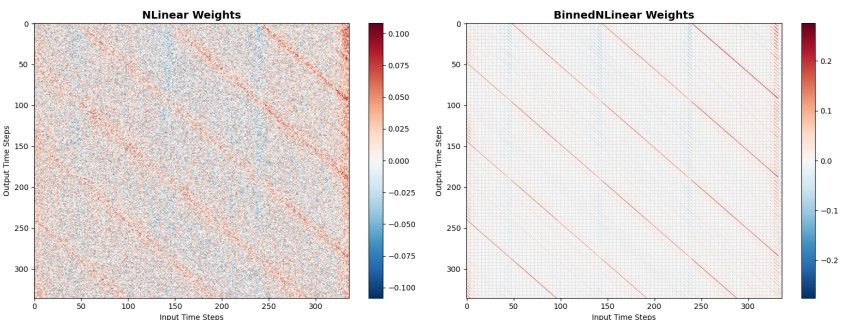

Figure 3: Model weights (ETTm2 dataset, horizon $T = 336$)

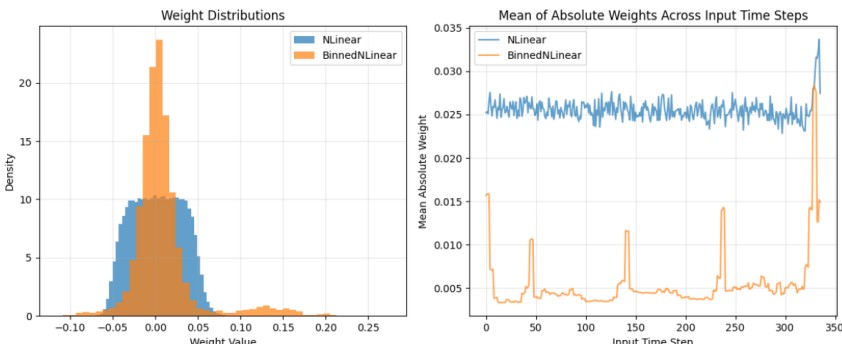

Figure 4: Model weight distribution and average across input time steps (ETTm2 dataset, horizon $T = 336$)

Figure 4 presents these weights from a different perspective. It can be seen that binning increases the spread of the weight values (can also be observed on a scale in Figure 3). At the same time, for the NLinear model, while fewer values are concentrated around 0, the overall distribution is more uniform and has less variability. The average weight across input time steps also reveals the results of different training strategies. For the binned model, the spikes are indicative as a result of the isolated and independent training. For the unbinned model, the mean absolute weights become noisy across the input series and potentially distorted, except for the spike of the short-term values of the input sequence. Several other examples are provided in the Appendix. Generally, the observations for other models are similar to the one presented above.

## 5.2 PARAMETER EFFICIENCY

**Theorem 3.** *Given a historical lookback window L, a forecast horizon H, and a constant periodicity w, binning reduces the total number of parameters of a linear model from LH to $\frac{LH}{w}$.*

The parameter reduction can be significant for a typical TSF setting. Suppose a lookback window and horizon of 720, and a constant periodicity of 24. Then the reduction of parameters is from 518,400 to 21,600 (96%). This suggests that exploiting periodic behavior is key in time series forecasting. With prediction error improvements, the results suggest that linear layers contain a substantial redundancy in parameters, which can be efficiently addressed through binning. Moreover, in a real-world setting, bins can be trained in parallel as their training is independent, providing computational efficiency.

## 5.3 APPLICABILITY

The empirical results presented show that for a particular dataset, the binning technique can perform consistently worse. This begs for an explanation: is this performance related to the data structure itself, and can a judgment be made whether binning will yield poorer predictions? The premise of

binning is to simplify or "decompose" the lookback relationships (which can include both trend and oscillations) into representable components. However, if the binned components do not provide a meaningful decomposition, the representations yielded can be poor. The observation in 7 is that the binned model makes poorer predictions on the periodic dips. This can suggest that the bin responsible for the predictions at those points potentially had a trend offset or some other distortion leading to improper learning.

The proposition is to conduct the experiments, which calculate the following metrics:

$$\text{Periodic Var \%} = \frac{1}{CN} \sum_{c=1}^{C} \sum_{n=1}^{N} \frac{\text{Var}(\hat{x}_{t_n:t_n+L-1}^{(c,n)})}{\text{Var}(x_{t_n:t_n+L-1}^{(c,n)})} \times 100\% \tag{5}$$

where $x_{t_n:t_n+L-1}^{(c,n)}$ represents a full sequence in a lookback window $n$ for channel $c$ and $\hat{x}_{t_n:t_n+L-1}^{(c,n)}$ represents the means of binned sequences (i.e., $\hat{x}_{t_n+i}^{(c,n)} = \mu_{i \bmod w}^{(c,n)}, \quad i = 0, 1, \ldots, L-1$) in a lookback window $n$ and channel $c$.

$$\text{Period CV} = \frac{1}{C} \sum_{c=1}^{C} \frac{\sigma(\{\text{PV}^{(c,n)}\}_{n=1}^{N})}{\mu(\{\text{PV}^{(c,n)}\}_{n=1}^{N})} \tag{6}$$

The interpretation of the metric is the comparison between the variation between bins and the variation within the sequence. The Coefficient of Variation (CV) represents the consistency of that periodic variation. For simplicity, the computations are carried out for the last 100 lookback windows of the training data.

Table 2: Temporal structure metrics for benchmarked datasets.

| Dataset | Period CV | Periodic Var% |
|---|---|---|
| ETTh1 | 0.050 | 56.41 |
| ETTh2 | 0.109 | 14.27 |
| Electricity | 0.010 | 82.54 |

Variance decomposition reveals that only 14.27% of ETTh2's variance originates from bin-specific means (the periodic component that binning isolates), compared to 56.41% for ETTh1. This weak periodic structure means the bins lack distinct characteristics to learn from. When binning trains separate models for each bin, it attempts to exploit a signal that explains less than 15% of the variance, while the remaining 85% (predominantly trend and noise) is distributed independently across bins. This fundamental weakness, combined with reduced temporal stability (i.e., larger Period CV), can potentially explain why binning fails for ETTh2. Broadly, this potentially suggests that binning is applicable to data with strong periodic characteristics.

### 5.4 PERIODICITY & BIN SELECTION

While selecting a number of bins equal to the periodicity is an intuitive choice, there is a potential premise for a different selection. For instance, the bins can be made twice as large in order to also capture the neighboring dynamics.

The results in 3 show an instance of a varying number of bins. They demonstrate that the number of bins equal to the periodicity provides the lowest error for periodic data.

For non-periodic data, the theoretical mechanism differs: binning may still provide benefits by creating structured subsampling that reduces temporal noise and parameter redundancy, even without perfect periodic decomposition. For practical purposes, it could be computed as a hyperparameter search.

### 5.5 LIMITATIONS & FUTURE WORK

This work provides results for non-periodic data. However, it is an open question why binning provides improvements for non-periodic data, being a subject to theoretical extensions and empirical

Table 3: Results for bin selection on benchmarked datasets. The **bold** indicates the best result and underline indicates the second best result in the respective column. The experiment was conducted for the ETTh1 dataset and the BinnedLinear model.

| Horizon | 96 | | 192 | | 336 | | 720 | |
|---|---|---|---|---|---|---|---|---|
| Metric | MSE | MAE | MSE | MAE | MSE | MAE | MSE | MAE |
| 1 Bin | 0.378 | 0.403 | 0.415 | 0.426 | 0.445 | 0.447 | 0.485 | 0.500 |
| 2 Bins | 0.378 | 0.396 | 0.412 | 0.419 | 0.443 | 0.442 | 0.479 | 0.495 |
| 4 Bins | 0.374 | 0.392 | 0.410 | 0.416 | 0.442 | 0.440 | 0.476 | 0.492 |
| 6 Bins | 0.373 | 0.391 | 0.410 | 0.415 | 0.442 | 0.438 | 0.472 | 0.489 |
| 12 Bins | 0.371 | 0.389 | 0.407 | 0.413 | 0.441 | 0.437 | 0.472 | 0.488 |
| 24 Bins | **0.369** | **0.387** | **0.405** | **0.411** | **0.437** | **0.434** | **0.470** | **0.486** |
| 48 Bins | 0.396 | 0.403 | 0.427 | 0.425 | 0.451 | 0.445 | 0.482 | 0.496 |

validation. Moreover, the bins do not have to be of equal size. There is potentially room for theoretical work, which can calculate the optimal number of bins based on the data properties. From the experimental results, it can be suggested that if binning does not work on one type of linear model for the dataset, it is highly unlikely it will yield better results for the other (i.e., the results for the ETTh2 dataset). Hence, there could be a further development in understanding what causes binned models to perform worse consistently on a certain type of data.

At the same time, a more direct approach could be to calculate the Mixture-of-Experts between models, similar to how the Mixture-of-Experts of linear models was done in Ni et al. (2024) work. This approach can combine models with a different number of bins into a single architecture and find an optimal combination. With that, there could also be ideas of overlapping bins, similar to sliding patches or chunks. However, this can drastically increase the number of required parameters and make the architecture complex. The computational and accuracy trade-offs would need to be considered. Additionally, investigation of binning across different channels can also be explored and studied.

## 6 CONCLUSION

This work contributes to the research of linear models in the TSF setting and explores binning as a potential that improves the capabilities of linear models for learning time series representations. The conceptual rationale lies behind the idea that binning can be an efficient way to learn time series data by leveraging periodic structure. Synthetic data experiments validate this finding, and the empirical data experiments also show improvements between binned and unbinned architectures, despite a substantial reduction in parameters. While this study provides fundamental and initial efforts of understanding the binning technique, there are a lot of possible extensions.

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

## A    APPENDIX

### A.1    INSPIRATION & ANALOGY

As mentioned in Section 1, the analogy comes from the energy sector. In the load forecasting setting, the reason for binning is the following: exposure to temperature at different hours can potentially distort modeling relationships. In Figure 5, it is evident that including relationships between temperature and load creates a scattered plot. Fitting a function, which will learn from this kind of relationship, is difficult. However, after selecting an hourly subset of the data, the relationship is much less scattered. Fitting a function, even with a lower number of parameters, is easier. This similar idea is translated to the sequential setting.

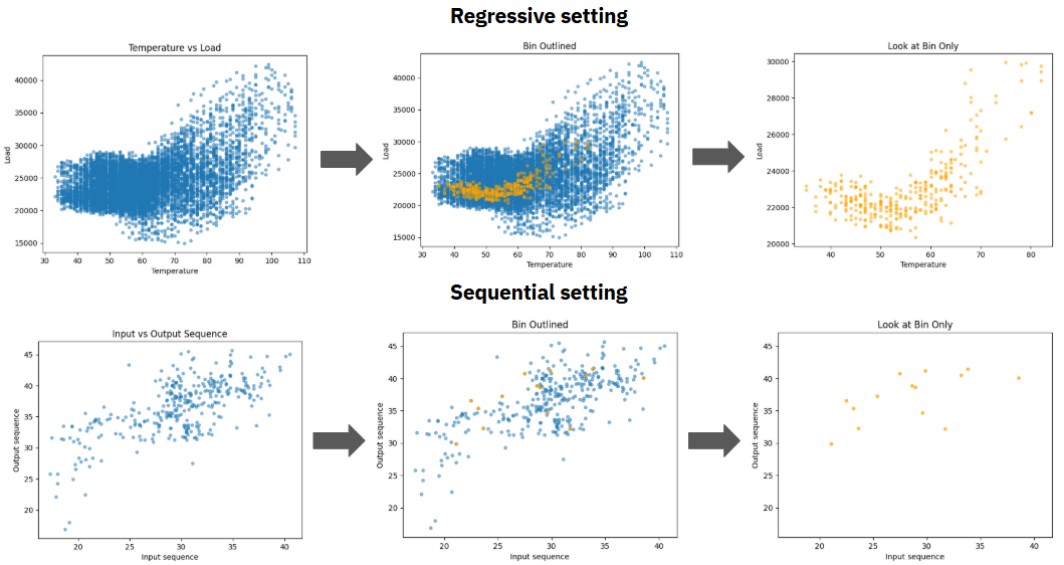

Figure 5: Representation of how binning processes data

## A.2 EXPERIMENTAL SET UP & FULL EMPIRICAL RESULTS

The code of this work is provided in the `github.com/Anonymous`.

### A.2.1 SYNTHETIC DATA EXPERIMENTS

The data is generated using Equation 3 with the addition of white noise $\mathcal{N}(0, 0.3)$ for 1000 points and $C_3 = 5$. For all iterations, $b = 10$, $C_1 = 1$ is used, and the value of $C_2$ equals to the TO ratio. The data is split into 50% training set and 50% test set.

An additional result for a high TO is presented below.

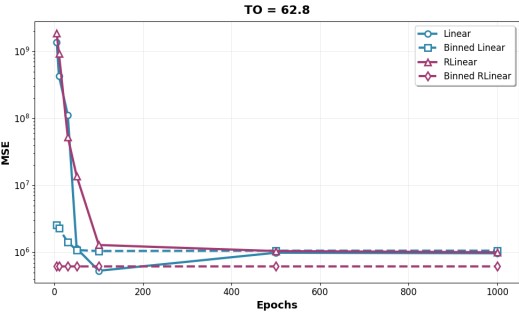

Figure 6: High TO result

### A.2.2 EMPRICIAL DATASET EXPERIMENTS

The codebase was adopted from the Lin et al. (2024b) GitHub repository. For ETTh1, ETTh2, ETTm1, and ETTm2, the lookback window was chosen to be 336, and for the Electricity dataset, the lookback was 720.

Full empirical results are provided in Tables 4 and 5.

Please note: the results provided were only simulated for seed 2023 of the framework due to computational and temporal limitations. Several of the experiments showed that binned models generally

do not show different results across different seeds (however, this is not always true for other models). This observation is left to be verified with iterations of other seeds as future work.

Table 4: Results on benchmarked datasets of the MSE metric. The **bold** indicates the best result and underline indicates the second best result in the respective column. The improvement indicates the improvement relative to the respective model without binning.

| Dataset | ETTh1 | | | | ETTh2 | | | | ETTm1 | | | | ETTm2 | | | | Electricity | | | |
|---|---|---|---|---|---|---|---|---|---|---|---|---|---|---|---|---|---|---|---|---|
| Horizon | 96 | 192 | 336 | 720 | 96 | 192 | 336 | 720 | 96 | 192 | 336 | 720 | 96 | 192 | 336 | 720 | 96 | 192 | 336 | 720 |
| Linear | 0.378 | 0.415 | 0.445 | 0.485 | 0.294 | 0.380 | 0.468 | 0.610 | 0.315 | 0.358 | 0.376 | 0.431 | 0.195 | 0.239 | 0.296 | 0.477 | 0.141 | 0.154 | 0.169 | 0.205 |
| Linear+Bin | 0.369 | 0.405 | 0.437 | 0.470 | 0.338 | 0.421 | 0.496 | 0.717 | 0.306 | 0.345 | 0.375 | 0.437 | 0.178 | 0.229 | 0.312 | 0.407 | 0.140 | 0.153 | 0.167 | 0.204 |
| Improv. | 0.009 | 0.010 | 0.008 | 0.005 | -0.044 | -0.041 | -0.028 | -0.107 | 0.009 | 0.013 | 0.001 | -0.006 | 0.017 | 0.010 | -0.016 | 0.070 | 0.001 | 0.001 | 0.002 | 0.001 |
| Improv. (%) | 2.38 | 2.41 | 1.80 | 1.03 | -14.97 | -10.79 | -5.98 | -17.54 | 2.86 | 3.63 | 0.27 | -1.39 | 8.72 | 4.18 | -5.41 | 14.68 | 0.71 | 0.65 | 1.18 | 0.49 |
| NLinear | 0.377 | 0.406 | 0.434 | 0.438 | **0.278** | **0.344** | **0.365** | **0.399** | **0.305** | 0.356 | 0.392 | 0.448 | **0.164** | 0.227 | 0.285 | 0.381 | **0.133** | **0.148** | **0.164** | 0.204 |
| NLinear+Bin | 0.369 | 0.406 | 0.433 | **0.443** | 0.301 | 0.351 | 0.369 | 0.428 | 0.317 | 0.346 | 0.379 | 0.441 | 0.171 | 0.222 | 0.276 | **0.370** | 0.140 | 0.153 | 0.168 | 0.207 |
| Improv. | 0.008 | 0 | 0.001 | -0.005 | -0.023 | -0.007 | -0.004 | -0.029 | -0.012 | 0.010 | 0.013 | 0.007 | -0.007 | 0.005 | 0.009 | 0.011 | -0.007 | -0.005 | -0.004 | -0.003 |
| Improv. (%) | 2.12 | 0 | 0.23 | -1.14 | -8.27 | -2.03 | -1.10 | -7.27 | -3.93 | 2.81 | 3.32 | 1.56 | -4.27 | 2.20 | 3.16 | 2.89 | -5.27 | -3.38 | -2.44 | -1.47 |
| DLinear | 0.378 | 0.410 | 0.442 | 0.485 | 0.315 | 0.387 | 0.467 | 0.712 | 0.309 | 0.343 | 0.376 | 0.432 | 0.183 | 0.255 | 0.319 | 0.487 | 0.133 | 0.162 | 0.177 | 0.212 |
| DLinear+Bin | 0.366 | 0.403 | 0.433 | 0.467 | 0.334 | 0.417 | 0.491 | 0.714 | 0.321 | 0.346 | 0.386 | 0.438 | 0.179 | 0.242 | 0.306 | 0.435 | 0.140 | 0.152 | 0.167 | 0.204 |
| Improv. | 0.012 | 0.007 | 0.009 | 0.018 | -0.019 | -0.020 | -0.024 | -0.002 | -0.012 | -0.003 | -0.010 | -0.006 | 0.004 | 0.013 | 0.013 | 0.052 | 0.007 | 0.010 | 0.010 | 0.008 |
| Improv. (%) | 3.17 | 1.71 | 2.04 | 3.71 | -6.03 | -5.17 | -5.14 | -0.28 | -3.88 | -0.87 | -2.66 | -1.39 | 2.19 | 5.10 | 4.08 | 10.68 | 5.27 | 6.17 | 5.65 | 3.78 |
| RLinear | 0.377 | 0.407 | 0.432 | 0.451 | 0.290 | 0.339 | 0.374 | 0.374 | 0.302 | 0.340 | 0.372 | 0.440 | 0.169 | 0.220 | 0.283 | 0.381 | 0.133 | 0.164 | 0.180 | 0.204 |
| RLinear+Bin | 0.371 | 0.407 | 0.434 | 0.460 | 0.301 | 0.355 | 0.376 | 0.428 | 0.307 | 0.345 | 0.374 | 0.430 | 0.169 | 0.223 | 0.282 | 0.376 | 0.140 | 0.153 | 0.169 | 0.208 |
| RLinear+Bin+G | 0.366 | 0.403 | 0.430 | 0.444 | 0.300 | 0.354 | 0.375 | 0.424 | 0.309 | 0.349 | 0.382 | 0.439 | 0.169 | 0.225 | 0.275 | 0.371 | 0.140 | 0.153 | 0.168 | 0.206 |
| Improv. | 0.011 | 0.004 | 0.002 | 0.007 | -0.010 | -0.015 | -0.001 | 0.008 | -0.005 | -0.005 | -0.005 | 0.010 | 0 | -0.003 | 0.008 | 0.010 | -0.007 | 0.011 | 0.012 | -0.002 |
| Improv. (%) | 2.92 | 0.98 | 0.46 | 1.55 | -3.45 | -4.42 | -0.27 | 1.85 | -1.66 | -1.47 | -1.34 | 2.27 | 0 | -1.36 | 2.83 | 2.62 | -5.27 | 6.71 | 6.67 | -0.98 |

Table 5: Results on benchmarked datasets of the MAE metric. The **bold** indicates the best result and underline indicates the second best result in the respective column. The improvement indicates the improvement relative to the respective model without binning.

| Dataset | ETTh1 | | | | ETTh2 | | | | ETTm1 | | | | ETTm2 | | | | Electricity | | | |
|---|---|---|---|---|---|---|---|---|---|---|---|---|---|---|---|---|---|---|---|---|
| Horizon | 96 | 192 | 336 | 720 | 96 | 192 | 336 | 720 | 96 | 192 | 336 | 720 | 96 | 192 | 336 | 720 | 96 | 192 | 336 | 720 |
| Linear | 0.403 | 0.426 | 0.447 | 0.500 | 0.357 | 0.414 | 0.471 | 0.554 | 0.362 | 0.395 | 0.393 | 0.427 | 0.298 | 0.321 | 0.354 | 0.476 | 0.243 | 0.254 | 0.271 | 0.306 |
| Linear+Bin | 0.387 | 0.411 | 0.434 | 0.486 | 0.391 | 0.444 | 0.493 | 0.606 | 0.349 | 0.375 | 0.393 | 0.432 | 0.274 | 0.306 | 0.374 | 0.427 | 0.235 | 0.248 | 0.263 | 0.297 |
| Improv. | 0.016 | 0.015 | 0.013 | 0.014 | -0.034 | -0.030 | -0.024 | -0.048 | 0.013 | 0.020 | 0 | -0.005 | 0.024 | 0.015 | -0.020 | 0.049 | 0.008 | 0.006 | 0.008 | 0.009 |
| Improv. (%) | 3.97 | 3.52 | 2.91 | 2.80 | -9.52 | -7.25 | -5.10 | -8.66 | 3.59 | 5.06 | 0 | -1.17 | 8.05 | 4.67 | -5.65 | 10.29 | 3.29 | 2.36 | 2.95 | 2.94 |
| NLinear | 0.398 | 0.414 | 0.432 | 0.453 | 0.339 | 0.382 | 0.404 | 0.438 | 0.347 | 0.383 | 0.403 | 0.435 | 0.254 | 0.295 | 0.339 | 0.395 | 0.229 | 0.242 | 0.259 | 0.292 |
| NLinear+Bin | 0.385 | 0.407 | 0.421 | 0.444 | 0.353 | 0.387 | 0.408 | 0.456 | 0.355 | 0.372 | 0.391 | 0.425 | 0.259 | 0.293 | 0.328 | 0.385 | 0.233 | 0.245 | 0.260 | 0.293 |
| Improv. | 0.013 | 0.007 | 0.011 | 0.009 | -0.014 | -0.005 | -0.004 | -0.018 | -0.008 | 0.011 | 0.012 | 0.010 | -0.005 | 0.002 | 0.011 | 0.010 | -0.004 | -0.003 | -0.001 | -0.001 |
| Improv. (%) | 3.26 | 1.69 | 2.55 | 1.99 | -4.13 | -1.30 | -0.99 | -4.11 | -2.31 | 2.87 | 2.98 | 2.30 | -1.97 | 0.68 | 3.24 | 2.53 | -1.75 | -1.24 | -0.39 | -0.34 |
| DLinear | 0.399 | 0.421 | 0.443 | 0.499 | 0.373 | 0.421 | 0.473 | 0.600 | 0.352 | 0.373 | 0.393 | 0.428 | 0.281 | 0.335 | 0.376 | 0.480 | 0.230 | 0.264 | 0.280 | 0.314 |
| DLinear+Bin | 0.385 | 0.409 | 0.432 | 0.485 | 0.388 | 0.441 | 0.487 | 0.604 | 0.357 | 0.376 | 0.402 | 0.433 | 0.274 | 0.320 | 0.364 | 0.442 | 0.234 | 0.246 | 0.262 | 0.295 |
| Improv. | 0.014 | 0.012 | 0.011 | 0.014 | -0.015 | -0.020 | -0.014 | -0.004 | -0.005 | -0.003 | -0.009 | -0.005 | 0.007 | 0.015 | 0.012 | 0.038 | -0.004 | 0.018 | 0.018 | 0.019 |
| Improv. (%) | 3.51 | 2.85 | 2.48 | 2.81 | -4.02 | -4.75 | -2.96 | -0.67 | -1.42 | -0.80 | -2.29 | -1.17 | 2.50 | 4.48 | 3.19 | 7.92 | -1.74 | 6.82 | 6.43 | 6.05 |
| RLinear | 0.399 | 0.415 | 0.430 | 0.462 | 0.348 | 0.384 | 0.413 | 0.454 | **0.344** | 0.367 | 0.384 | 0.427 | 0.255 | 0.290 | 0.337 | 0.396 | **0.228** | 0.265 | 0.280 | 0.292 |
| RLinear+Bin | 0.386 | 0.407 | 0.420 | 0.444 | 0.355 | 0.394 | 0.415 | 0.449 | 0.348 | 0.371 | 0.384 | 0.418 | 0.257 | 0.293 | 0.332 | 0.390 | 0.233 | 0.244 | 0.260 | 0.293 |
| RLinear+Bin+G | 0.385 | 0.408 | 0.425 | 0.456 | 0.355 | 0.395 | 0.417 | 0.451 | 0.350 | 0.373 | 0.394 | 0.425 | 0.257 | 0.296 | 0.328 | 0.388 | 0.234 | 0.245 | 0.261 | 0.293 |
| Improv. | 0.014 | 0.008 | 0.010 | 0.017 | -0.007 | -0.010 | -0.002 | 0.005 | -0.004 | -0.004 | -0.002 | 0.009 | -0.002 | -0.003 | 0.009 | 0.008 | -0.005 | 0.021 | 0.020 | -0.001 |
| Improv. (%) | 3.51 | 1.93 | 2.33 | 3.68 | -2.01 | -2.60 | -0.48 | 1.10 | -1.16 | -1.09 | -0.52 | 2.11 | -0.78 | -1.03 | 2.67 | 2.02 | -2.19 | 7.92 | 7.14 | -0.34 |

### A.2.3 COMPARISON WITH OTHER MODELS

A crude comparison with other models in the literature is conducted. The results of other models from (Lin et al., 2025) are presented below. Please note that the comparison is crude, as that work selects a lookback window of 720 for all models, while a lookback window of 336 is selected for the majority of the tested datasets in this work. The model of comparison chosen is the Global Binned RLinear model.

Table 6: Results on benchmarked datasets of the MSE metric. The **bold** indicates the best result and underline indicates the second best result in the respective column. The improvement indicates the improvement relative to the respective model without binning.

| Dataset | ETTh1 | | | | ETTh2 | | | | Electricity | | | |
|---|---|---|---|---|---|---|---|---|---|---|---|---|
| Horizon | 96 | 192 | 336 | 720 | 96 | 192 | 336 | 720 | 96 | 192 | 336 | 720 |
| Autoformer | 0.594 | 0.606 | 0.618 | 0.742 | 0.471 | 0.485 | 0.474 | 0.512 | 0.208 | 0.229 | 0.239 | 0.235 |
| iTransformer | 0.396 | 0.430 | 0.480 | 0.700 | 0.311 | 0.392 | 0.415 | 0.425 | 0.133 | 0.152 | 0.169 | **0.194** |
| PatchTST | 0.375 | 0.418 | 0.462 | 0.495 | 0.290 | 0.382 | 0.421 | 0.404 | **0.130** | **0.147** | **0.162** | 0.201 |
| FEDformer | 0.439 | 0.469 | 0.521 | 0.624 | 0.398 | 0.412 | 0.440 | 0.469 | 0.255 | 0.288 | 0.310 | 0.323 |
| FITS | 0.382 | 0.417 | 0.436 | 0.433 | **0.272** | **0.333** | **0.355** | **0.378** | 0.145 | 0.159 | 0.175 | 0.212 |
| SparseTSF (Linear) | **0.362** | 0.404 | 0.435 | **0.426** | 0.294 | 0.340 | 0.360 | 0.383 | 0.139 | 0.151 | 0.166 | 0.207 |
| RLinear+Bin+G | 0.366 | **0.403** | **0.430** | 0.444 | 0.300 | 0.354 | 0.375 | 0.424 | 0.140 | 0.153 | 0.168 | 0.206 |

Table 7: Results on benchmarked datasets of the MAE metric. The **bold** indicates the best result and underline indicates the second best result in the respective column. The improvement indicates the improvement relative to the respective model without binning.

| Dataset | ETTh1 | | | | ETTh2 | | | | Electricity | | | |
|---|---|---|---|---|---|---|---|---|---|---|---|---|
| Horizon | 96 | 192 | 336 | 720 | 96 | 192 | 336 | 720 | 96 | 192 | 336 | 720 |
| Autoformer | 0.584 | 0.588 | 0.595 | 0.660 | 0.487 | 0.492 | 0.488 | 0.513 | 0.323 | 0.344 | 0.358 | 0.349 |
| iTransformer | 0.426 | 0.450 | 0.486 | 0.608 | 0.363 | 0.414 | 0.438 | 0.455 | 0.229 | 0.249 | 0.266 | 0.288 |
| PatchTST | 0.404 | 0.430 | 0.460 | 0.497 | 0.354 | 0.419 | 0.448 | 0.444 | **0.225** | **0.241** | **0.257** | **0.266** |
| FEDformer | 0.479 | 0.493 | 0.528 | 0.587 | 0.449 | 0.457 | 0.474 | 0.506 | 0.362 | 0.387 | 0.405 | 0.410 |
| FITS | 0.405 | 0.425 | 0.442 | 0.455 | **0.336** | **0.375** | **0.396** | **0.423** | 0.248 | 0.260 | 0.275 | 0.305 |
| SparseTSF (Linear) | 0.389 | 0.412 | 0.428 | **0.448** | 0.346 | 0.377 | 0.398 | 0.425 | 0.234 | 0.245 | 0.260 | 0.297 |
| RLinear+Bin+G | **0.385** | **0.408** | **0.425** | 0.456 | 0.355 | 0.395 | 0.417 | 0.451 | 0.234 | 0.245 | 0.261 | 0.293 |

The results show that one of the proposed models showcases excellent performance on the ETTh1 dataset, beating the other models proposed in the literature across almost all horizons. For the ETTh2 dataset, the model is not competitive, as binned models do not provide improvements. For the Electricity dataset, the model falls short to some of the transformer models.

### A.3    DISCUSSION ON ETTH2 DATASET

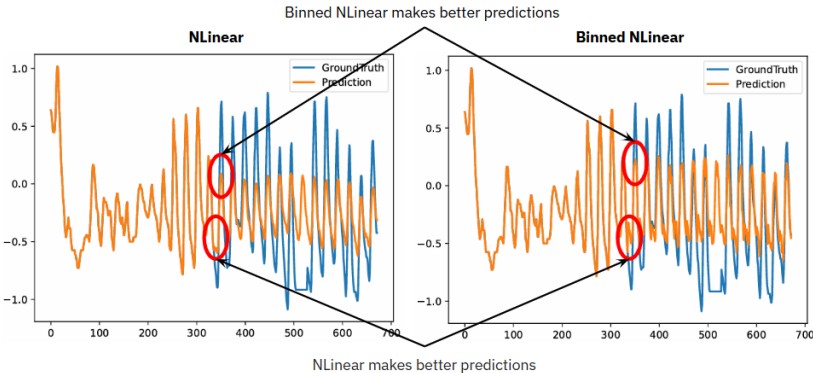

Figure 7: Comparison of predictions for the ETTh2 dataset

The visual is presented in Figure 7. The observation is that the NLinear model makes smoother drop predictions, while the binned model has some kind of bump during this period. However, the binned model more accurately predicts the amplitude of the spikes. The drop bumps become a large source of error, hence making the model performance worse than its unbinned counterpart. As mentioned in Section 5, this can show a potential for the mixture of models. If the combination that uses a binned model for spikes and an unbinned model for drops can be obtained, the overall predictions can be more accurate.

## A.4   OTHER REPRESENTATION EXAMPLES

### A.4.1   LINEAR MODEL

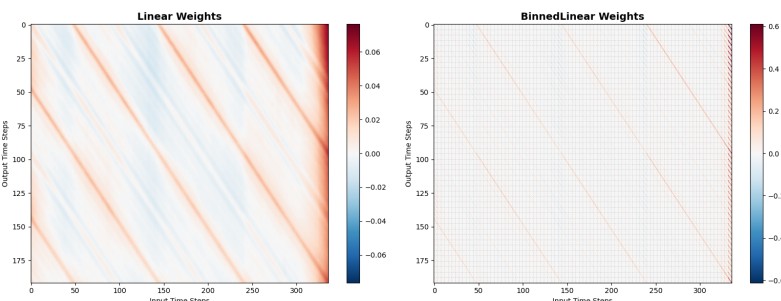

Figure 8: Model weights (ETTm2 dataset, horizon $T = 192$)

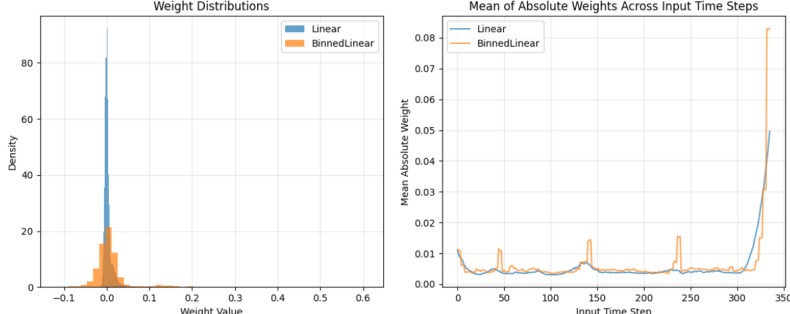

Figure 9: Model weight distribution and average across input time steps (ETTm2 dataset, horizon $T = 192$)

### A.4.2   DLINEAR MODEL

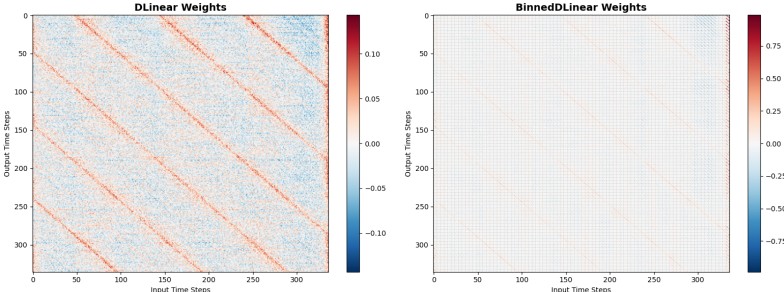

Figure 10: Model weights (ETTm2 dataset, horizon $T = 336$)

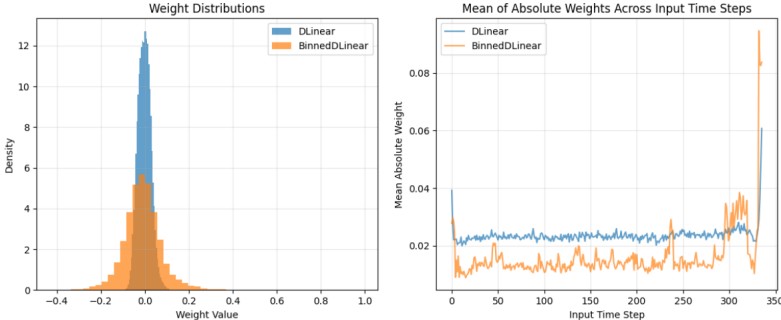

Figure 11: Model weight distribution and average across input time steps (ETTm2 dataset, horizon $T = 336$)

### A.4.3 RLINEAR MODEL

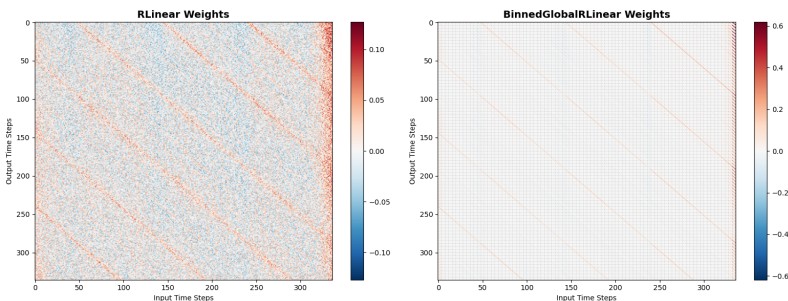

Figure 12: Model weights (ETTm2 dataset, horizon $T = 336$)

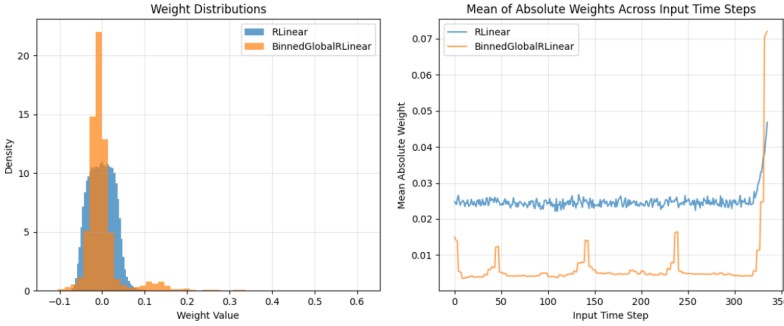

Figure 13: Model weight distribution and average across input time steps (ETTm2 dataset, horizon $T = 336$)

## A.5 DEFERRED PROOFS

*Proof.* **Theorem 1** The time series is of the form:

$$x(t) = s(t) + p(t)$$

where $s(t)$ is a periodic component with period $w$, i.e., $s(t + w) = s(t)$ for all $t$, and $p(t)$ is a trend component.

Define bin $b \in \{0, 1, \ldots, w - 1\}$ to include all time indices where $t \equiv b \pmod{w}$. Then, any time index in bin $b$ is:

$$t = kw + b, \quad \text{for } k \in \mathbb{Z}_{\geq 0}.$$

For all such $t$:
$$x(t) = x(kw + b) = s(kw + b) + p(kw + b)$$

Since $s(t)$ is periodic with period $w$:
$$s(kw + b) = s(b), \quad \text{for all } k$$

Thus, the binned time series for bin $b$:
$$x^b(t) = x(kw + b) = s(b) + p(kw + b)$$

Denote $c_b = s(b)$ a constant depending only on the bin and $p^b(t) = p(kw + b)$ the trend sampled at period offset $b$.

Then:
$$x^b(t) = c_b + p^b(k)$$
which shows that the seasonal component has been reduced to a constant offset $c_b$, and the remaining variation over $k$ is due to the trend $p(t)$.

Hence, binning by period $w$ removes the seasonal variation within each bin, and the resulting binned series reflects only the trend component with a bin-specific intercept. $\qquad \square$

*Proof.* **Theorem 2** By definition, for each bin $b$ and time index $t = kw + b$:
$$x^b(t) = x(kw + b) = s(kw + b) \cdot p_1(kw + b) + p_2(kw + b)$$

Since $s(t)$ is periodic with period $w$, it follows that:
$$s(kw + b) = s(b) = c_b$$
which is constant for fixed $b$ and all $k$.

Define:
$$p_1^b(t) = p_1(kw + b), \quad p_2^b(t) = p_2(kw + b)$$
which are the trend components sampled at the phase defined by bin $b$.

Substituting:
$$x^b(t) = c_b \cdot p_1^b(t) + p_2^b(t)$$

This shows that for each bin $b$, the seasonal component $s(t)$ becomes a constant multiplicative factor $c_b$, and the time series reflects a bin-specific trend profile composed of a scaled version of $p_1(t)$ and an additive component $p_2(t)$, both sampled at offsets of $b$.

Therefore, binning eliminates the time-varying seasonal component within each bin and leaves a trend signal whose shape and intercept vary across bins. $\qquad \square$

**Theorem 2 (Extended).** *Let the time series be given by:*
$$x(t) = \prod_{i=1}^{n} s_i(t) \cdot p_1(t) + p_2(t) \tag{7}$$

*where each $s_i(t)$ is a periodic function with period $w_i$ (i.e., $s_i(t + w_i) = s_i(t)$ for all t), and $p_1(t)$ and $p_2(t)$ are trend components. Let $w = lcm(w_1, w_2, \ldots, w_n)$ denote the least common multiple of all periods. Then, for each bin $b$, the binned time series has the form:*
$$x^b(t) = c_b \cdot p_1^b(t) + p_2^b(t) \tag{8}$$

*where $c_b = \prod_{i=1}^{n} s_i(b)$ is a constant, and $p_1^b(t) = p_1(kw+b)$ and $p_2^b(t) = p_2(kw+b)$ are resampled versions of the trend components.*

*Proof.* By definition, for each bin $b$ and time index $t = kw + b$:
$$x^b(t) = x(kw + b) = \prod_{i=1}^{n} s_i(kw + b) \cdot p_1(kw + b) + p_2(kw + b) \tag{9}$$

Since $w = \text{lcm}(w_1, w_2, \ldots, w_n)$, for each $i \in \{1, 2, \ldots, n\}$ there exists an integer $m_i$ such that $w = m_i \cdot w_i$. Therefore:

$$kw = k \cdot m_i \cdot w_i \tag{10}$$

Since each $s_i(t)$ is periodic with period $w_i$, it follows that:

$$s_i(kw + b) = s_i(k \cdot m_i \cdot w_i + b) = s_i(b) \tag{11}$$

which is constant for fixed $b$ and all $k$.

Define:

$$c_b = \prod_{i=1}^{n} s_i(b) \tag{12}$$

which is constant for each bin $b$.

Define:

$$p_1^b(t) = p_1(kw + b), \quad p_2^b(t) = p_2(kw + b) \tag{13}$$

which are the trend components sampled at the phase defined by bin $b$.

Substituting into the original equation:

$$x^b(t) = c_b \cdot p_1^b(t) + p_2^b(t) \tag{14}$$

This shows that for each bin $b$, all seasonal components $s_1(t), s_2(t), \ldots, s_n(t)$ collapse into a single constant multiplicative factor $c_b$, and the time series reflects a bin-specific trend profile composed of a scaled version of $p_1(t)$ and an additive component $p_2(t)$, both sampled at offsets of $b$.

Therefore, binning at the common period $w$ eliminates all time-varying seasonal components within each bin and leaves a trend signal whose shape and intercept vary across bins. $\square$

**Remark 1 (Commensurate Periods).** The extended theorem requires that all periods $w_1, w_2, \ldots, w_n$ be *commensurate*, meaning their least common multiple $w = \text{lcm}(w_1, w_2, \ldots, w_n)$ exists and is finite. This occurs when all period ratios $w_i/w_j$ are rational numbers.

If the periods are *incommensurate* (i.e., at least one ratio $w_i/w_j$ is irrational), then no finite common period exists, and binning at any single period $w$ will not simultaneously eliminate all periodic components. In such cases, at least one seasonal component $s_i(t)$ will remain time-varying within each bin, and the simplification to the form $x^b(t) = c_b \cdot p_1^b(t) + p_2^b(t)$ does not hold.

In practical applications, if data exhibits multiple periodicities, one could verify their commensurability through spectral analysis (e.g., FFT) before applying the binning technique. For approximately commensurate periods arising from real-world measurements, the approximation error should be evaluated.

*Proof.* **Theorem 3** Each bin has $\frac{L}{w}$ input and $\frac{H}{w}$ output, requiring $\frac{L}{w} \times \frac{H}{w}$ parameters. With $w$ bins, the total number of parameters becomes $w \times \frac{L}{w} \times \frac{H}{w}$ . $\square$

## A.6 USAGE OF LLMS

This work was done with the assistance of LLMs in the following way: 1) coding support (Claude Opus 4.1); 2) writing polishing (Claude Sonnet 4 & OpenAI GPT 5). LLMs were **not** used for research ideation, retrieval, or discovery. The authors are responsible for all the mistakes.

