# OpenReview forum: "Binning of Linear Models for Time Series Forecasting"
_ICLR.cc/2026/Conference — Submitted to ICLR 2026_

### Official Review · Reviewer_mKLw · 2025-10-25

**Soundness:** 2
**Presentation:** 3
**Contribution:** 2
**Rating:** 4
**Confidence:** 3

**Summary:**

This paper introduces a binning technique for time series forecasting that partitions the lookback window based on periodicity rather than temporal adjacency. Each bin contains data points that are w time steps apart (e.g., all same-hour observations for hourly data with daily period), and is processed independently by a separate linear layer. The authors provide theoretical analysis showing that binning eliminates seasonal variation within bins and reduces parameters from LH to LH/w. The technique is validated on synthetic data and five benchmark datasets (ETTh1, ETTh2, ETTm1, ETTm2, Electricity) using four linear model variants.

**Strengths:**

The periodicity-based binning technique they provided sounds interesting and novel. Unlike some previous value-based discretization (e.g., https://arxiv.org/pdf/2005.10111), this work groups data by temporal phase within periods, which is conceptually intuitive for seasonal time series.
They provided strong theoretical foundations for their method as well as empirical validation in both synthetic datasets and real benchmarks.
They made a comprehensive analysis for each experiment, especially the weight visualization, which provides useful insights.

**Weaknesses:**

1. Although the binning technique in this work is novel (as far as I know), there were some other related works (like https://arxiv.org/pdf/2005.10111) that also use the 'binning' concept in a different way. The authors should make more comparisons with such related works.
2. They made some theoretical claims that were not thoroughly validated in experiments. For instance, they claimed a significant parameter efficiency ($LH \to LH/w$), but why is it not reflected in the experiments? Did the binning use less parameters compared to the baselines? Also, they claimed the method can be applied to any model, but they only did experiments on linear ones.
3. They made strong assumptions in their theoretical part, for example they assumed there is only one single period and it is perfectly periodic throughout the entire time series, which should be more carefully dealt with (for example in AutoFormer they only required approximate periods and learns it). Also they assumes the period $w$ is known, which is acceptable in real cases but should be more clearly stated.
4. Their method consistently fails in Etth2 (and for some models in some other datasets). They made a superficial discussion in the Appendix but it would be better if they can analyze deeper, for example why binning fails here and in what cases should be use binning.
5. More ablation studies on experiment hyperparameters should be done, and they should do more comparisons with other methods instead of only the linear models themselves.

**Questions:**

I think there are many interesting ideas in this paper which are underdeveloped, and there are some other concerns. If the author manage to address some of these concerns I think this would significantly improve the quality of this paper (and of course I'd considering giving higher score).
1. As mentioned in 'weaknesses', was the parameter efficiency reflected in your experiments? If not, why not?
2. Can you test the method on non-linear models? Is it also effective?
3. Can you explain what may happen if the assumptions are slightly imperfect? For example what will hapen if you got a wrong $w$? How can you handle if their are multiple seasonalities?
4. You only tested the $b=w$ case but left other cases as future works. Can you briefly justify this (for example why not $b=w/2$ or $b=2w$) either theoretically or empirically?
5. As mentioned in 'weaknesses', the binned models consistently fail on ETTh2 across all model types, but only a superficial discussion was made, which is unsatisfactory. Have you investigated why? Does ETTh2 violate your periodicity assumptions? Would using a different period w or number of bins b help? Understanding this failure would strengthen the paper by clarifying when binning should/shouldn't be used.
6. Section 2 discusses SparseTSF in detail and notes it also uses periodic sampling, but you never provide a direct empirical comparison. Could you include SparseTSF results in Table 1 to show the marginal contribution of your binning approach compared to this closely related method?
7. Tables 1-2 show that global relative binning (grb) generally outperforms local absolute binning (lab), but the reason is not clearly explained. Is this because grb provides better scale normalization, or because it allows better cross-series learning? Could you provide more analysis on when each strategy is preferable?
8. Figure 2 (synthetic experiments) shows binned models perform best at high TO ratios (oscillation-dominated). Can you characterize which real datasets have high vs. low TO ratios? Does this explain why binning works well on some datasets (ETTm2) but fails on others (ETTh2)?

---

> ### Author Response · Authors · 2025-11-22
>
> Thank you for the review and comments. Below are the answers to the questions and comments.
>
> 1) Thank you for the reference to https://arxiv.org/pdf/2005.10111. I reviewed the work and added the following in the Introduction Section:
>
> "One form of binning, denoted as discretization, is proposed in (Rabanser et al., 2020). The approach proposed primarily uses value-based quantile binning as a normalization technique for deep learning models. This work introduces period-aligned binning for linear models, where bins correspond to specific phases within temporal cycles rather than value ranges."
>
> 2) I hope to compute parameter efficiency metrics following the methodology in SparseTSF (Table 3 of their paper), providing a complete comparison table before the discussion deadline. Preliminary observations show ~30s epoch time (comparable to SparseTSF's 31.3s).
>
> 3) The binning framework is theoretically model-agnostic (f(·) in Algorithm 1 can be any architecture). I deliberately started with linear models for interpretable validation of the theoretical framework (such as weight analysis in Section 5.1). Implementing transformer-based models is a natural extension, and (if time permits) I hope to provide some of the iterations.
>
> 4) I have added bin ablation results (see response to Reviewer LqZz and Section 5.4) showing graceful degradation (i.e., if one chooses a wrong $w$, you will have a worse performance). Performance peaks at true periodicity (24 for ETTh1) and degrades at 48 bins. I have also extended Theorem 2 (Appendix) to handle commensurate multi-periodic data (also in response to Reviewer LqZz).
>
> 5) Theoretically, Theorem 1 shows that when $b=w$, each bin captures exactly one phase of the period, causing seasonal components to collapse to constants. If $b=w/2$, each bin would capture two phases, leaving seasonal variation within bins. If $b=2w$, we oversample and do not have enough data per bin. Empirically, this is shown with the bin iteration results (added in Section 5.4).
>
> 6) I have added some of the metrics to quantify a proxy for the TO ratio (Section 5.3):
>
> $$
> PeriodicVar =
> \frac{1}{C N}
> \sum_{c=1}^{C}\sum_{n=1}^{N}
> \frac{Var(xh_{c,n})}{Var(x_{c,n})}
> $$
>
> (variance of binned means over sequence variance)
>
> $$
> PeriodCV =
> \frac{1}{C}
> \Big(
> \sum_{c=1}^{C}
> \frac{ \sigma(PV_{c,n}) }{ \mu(PV_{c,n}) }
> \Big)
> $$
>
> The idea is that one can measure the variation between bins and the variation within the sequence. The Coefficient of Variation (CV) represents the consistency of that periodic variation. The results are:
>
> | Dataset     | Period CV | Periodic Var% |
> |-------------|-----------|----------------|
> | ETTh1 | 0.050 | 56.41  |
> | ETTh2 | 0.109 | 14.27  |
> | Electricity | 0.010 | 82.54   |
>
> The suggestion is that if the CV is high and the Periodic Variance is low, the binning is likely not to perform well.
>
> 7) I have added a comparison Table with SparseTSF (and other models) in the Appendix.
>
> 8) I have added a footnote clarifying:
>
> "This assumption resonates with some of the discussion made in (Lin et al., 2024b). If the data are periodic,
> periodicity can potentially be identified through the ACF function, and for certain domains, the periodicity
> might be known. For other cases and practical purposes, $w$ can be treated as a hyperparameter."
>
> I agree that AutoFormer's learned approximate periods are more flexible - this is a valid limitation of the current approach.
>
> 9) This is a great point, and I am considering adding additional experiments on this (for other models). My intuition is that global normalization most likely provides a better normalization.

---

> > ### Comment · Reviewer_mKLw · 2025-11-27
> >
> > I appreciate the effort, and some of my concerns are addressed, but some of them still are not.  The authors have managed to justify some claims they made in the paper, but I think the proven scope of the method looks quite limited: the method is useful in limited cases: empirically only for linear models, and has some strong constraints on the properties of the time series. The efficiency is also not a significant improvement compared to SparseTSF (~30s against 31.3s). I would wait for further evidence to decide if I would adjust the score.

---

### Official Review · Reviewer_LqZz · 2025-10-27

**Soundness:** 1
**Presentation:** 3
**Contribution:** 2
**Rating:** 2
**Confidence:** 4

**Summary:**

In the paper under review, a binning method for linear models for time series forecasting is proposed, where the binning is used to capture the periodic / seasonal part of the time series.

**Strengths:**

1) The paper is well written and easy to follow.
2) The approach is simple and can be applied to any linear forecasting model.

**Weaknesses:**

1) The paper makes strong assumptions about the structure of the time series. It is assumed that the (original) time series model decomposes in a seasonal and a trend components. This only reflects a small subset of time series models.

2) The paper leaves many aspects unclear. What is the purpose of the bin function f(.)? Is it implicitly assumed that w < L holds? How are f, b, and w selected in the experiments?

3) The experiments are limited. The used datasets are rather simple with a low sample frequency. There is data with high frequency from for instance the manufacturing domain, e.g., RUL forecasting. Further, the benefits of the binning are very mixed.

4) Overall the contributions of this paper are too limited. It feels more like a best practice addon to linear models, which should be better presented as workshop contributions rather than contributions to a large AI/ML conference.

**Questions:**

Can the binning also applied to nonlinear models like transformers?

What is the purpose of the bin function f(.)? Is it implicitly assumed that w < L holds? How are f, b, and w selected in the experiments?

**Details Of Ethics Concerns:**

There are no ethical concerns.

---

> ### Author Response · Authors · 2025-11-22
>
> Thank you for the review and comments. Below are the answers to the questions and comments.
>
> 1) I assume that the time series has seasonal and trend components, but I actually try to argue that binning can be applied to a more sophisticated representation of time series from a theoretical perspective. This representation is not necessarily tailored to the practical applications, but is rather used to propose a conceptual rationale. Because this representation is limited, the experiments on real-world datasets are conducted.
>
> 2) The binning framework is model-agnostic. The function f(·) can represent any architecture - linear, transformer, etc. In this work, I deliberately chose linear models for initial validation because they provide an interpretable analysis of learned representations (Section 5.1), which would be difficult with more complex architectures. Extending to transformers is a possible next step.
>
> 3) In this work, f(.) is selected to be linear models [all the models iterated in the results table can be treated as f(.)]. I added the following footnote regarding $w$:
>
> "This assumption [$w=b$] resonates with some of the discussion made in (Lin et al., 2024b) [https://arxiv.org/pdf/2405.00946]. If data are periodic,
> periodicity can potentially be identified through the ACF function, and for certain domains, the periodicity
> might be known. For other cases and practical purposes, $w$ can be treated as a hyperparameter."
>
> Additionally, I added a traversal across different numbers of bins for the Binned Linear model for the ETTh1 dataset. The table is provided below:
>
> | **Bins** | **MSE (96)** | **MAE (96)** | **MSE (192)** | **MAE (192)** | **MSE (336)** | **MAE (336)** | **MSE (720)** | **MAE (720)** |
> | -------- | ------------ | ------------ | ------------- | ------------- | ------------- | ------------- | ------------- | ------------- |
> | 1 Bin    | 0.378        | 0.403        | 0.415         | 0.426         | 0.445         | 0.447         | 0.485         | 0.500         |
> | 2 Bins   | 0.378        | 0.396        | 0.412         | 0.419         | 0.443         | 0.442         | 0.479         | 0.495         |
> | 4 Bins   | 0.374        | 0.392        | 0.410         | 0.416         | 0.442         | 0.440         | 0.476         | 0.492         |
> | 6 Bins   | 0.373        | 0.391        | 0.410         | 0.415         | 0.442         | 0.438         | 0.472         | 0.489         |
> | 12 Bins  | 0.371 | 0.389 | 0.407  | 0.413  | 0.441  | 0.437  | 0.472  | 0.488  |
> | 24 Bins  | **0.369**    | **0.387**    | **0.405**     | **0.411**     | **0.437**     | **0.434**     | **0.470**     | **0.486**     |
> | 48 Bins  | 0.396        | 0.403        | 0.427         | 0.425         | 0.451         | 0.445         | 0.482         | 0.496         |
>
> 4) Yes, it is implicitly assumed that $w<L$. From the practical standpoint, the lookback window usually captures all the periods within the input. If periodicity exceeded the lookback window length, then it would be practical to extend the lookback window to capture all periods in training.
>
> 5) I agree that this work could be extended to more models beyond linear and datasets (I am currently working on more experiments and hoping to add them soon). The idea was to introduce a novel decomposition perspective for time series forecasting that differs from existing approaches. While the empirical validation focuses on linear models and standard benchmarks, the core contribution is conceptual and theoretical. Theorems (with the multiperiodic extension, added in the Appendix) establish a formal foundation showing how binning transforms the learning problem. It's a principled decomposition that isolates periodic structure from trend dynamics. The weight analysis reveals that binning fundamentally changes what models learn—creating sparse, interpretable representations versus dense, potentially noisy ones. This insight about parameter redundancy (96% reduction) challenges assumptions about how temporal models should be structured. Regarding the mixed results, I added the following metrics:
>
> $$
> PeriodicVar =
> \frac{1}{C N}
> \sum_{c=1}^{C}\sum_{n=1}^{N}
> \frac{Var(xh_{c,n})}{Var(x_{c,n})}
> $$
>
> (variance of binned means over sequence variance)
>
> $$
> PeriodCV =
> \frac{1}{C}
> \Big(
> \sum_{c=1}^{C}
> \frac{ \sigma(PV_{c,n}) }{ \mu(PV_{c,n}) }
> \Big)
> $$
>
> The results are:
>
> | Dataset     | Period CV | Periodic Var% |
> |-------------|-----------|----------------|
> | ETTh1 | 0.050 | 56.41  |
> | ETTh2 | 0.109 | 14.27  |
> | Electricity | 0.010 | 82.54   |
>
> This weak periodic structure means the bins lack distinct characteristics to learn from. This can potentially provide a way to predict when binning will succeed.

---

### Official Review · Reviewer_zRHK · 2025-10-31

**Soundness:** 2
**Presentation:** 2
**Contribution:** 2
**Rating:** 2
**Confidence:** 4

**Summary:**

The paper explores the idea of applying binning for time-series forecasting.

**Strengths:**

The perspective is somewhat novel and provides an interesting way to think about different way of representing temporal periodicity.

**Weaknesses:**

**Major Comments**

1. The assumption in Line 113 is too strong and impractical. How can one know the period $w$ in advance and set the number of bins accordingly? Moreover, how would the method handle time series that exhibit multiple or varying periods?

2. In the methodology section, it is unclear what it means to “stack” the predictions of different bins. Does this simply mean summing them, or is there another aggregation mechanism?

3. The experimental evaluation is very limited. Many recent and representative time-series forecasting models after DLinear and RLinear are missing from the baselines. In addition, the experiments use only ETT and Electricity datasets, which makes the evaluation insufficiently comprehensive.

4. The paper does not explain how the number of bins is determined. Furthermore, it would be important to show how performance changes as the number of bins varies.

**Minor Comments**
5. The paper should at least cite the baseline models NLinear, DLinear, and RLinear.

**Questions:**

1. Clarify the assumption regarding the period ($w$).
Does the method assume that the period is known in advance, or does it use an arbitrary or learned value? Please make this explicit and discuss how the approach handles varying or unknown periods. (W1, W4)

2. Clarify the meaning of “stacking” in the method.
It is unclear what “stack” precisely means in this context - does it refer to summing, concatenation, or another form of aggregation? Please specify this in the methodology section. (W2)

3. Expand and strengthen the experimental evaluation.
Consider including more recent and representative baselines as well as additional datasets to demonstrate the robustness and generalizability of the proposed approach. (W3)

---

> ### Author Response · Authors · 2025-11-22
>
> Thank you for the review and comments. Below are the answers to the questions and comments.
>
> 1) The period $w$ can be calculated by an Auto Correlation Function, which would show the strongest relationship with the periodic lag. This resonates with the approach taken in the SparseTSF paper (https://arxiv.org/pdf/2405.00946), which also leverages the periodicity of the data. For certain domains, the period can be assumed to be known in advance, and for practical applications, $w$ can also be treated as a hyperparameter. I added a footnote discussing this:
>
> "This assumption resonates with some of the discussion made in (Lin et al., 2024b). If the data are periodic,
> periodicity can potentially be identified through the ACF function, and for certain domains, the periodicity
> might be known. For other cases and practical purposes, $w$ can be treated as a hyperparameter."
>
> For multiperiodic data, I have extended Theorem 2 to show that binning provides meaningful decomposition when periods are commensurate (see Appendix for full proof):
>
> **Theorem 2 (Extended).** *Let the time series be given by:*
> \begin{equation}
> x(t) = \prod_{i=1}^{n} s_i(t) \cdot p_1(t) + p_2(t)
> \end{equation}
> *where each $s_i(t)$ is a periodic function with period $w_i$ (i.e., $s_i(t + w_i) = s_i(t)$ for all $t$), and $p_1(t)$ and $p_2(t)$ are trend components. Let $w = \text{lcm}(w_1, w_2, \ldots, w_n)$ denote the least common multiple of all periods. Then, for each bin $b$, the binned time series has the form:*
> \begin{equation}
> x^b(t) = c_b \cdot p^b_1(t) + p^b_2(t)
> \end{equation}
> *where $c_b = \prod_{i=1}^{n} s_i(b)$ is a constant, and $p^b_1(t) = p_1(kw + b)$ and $p^b_2(t) = p_2(kw + b)$ are resampled versions of the trend components.*
>
> Additionally, I should clarify that the ETTm1 and ETTm2 datasets are not periodic, yet binning still demonstrates improvements. For these data, the theoretical mechanism differs: binning may still provide benefits by creating structured subsampling that reduces temporal noise and parameter redundancy (Theorem 3), even without perfect periodic decomposition. For similar cases, $w$ would be iterated.
>
> 2) By stacking, I mean concatenation in a single prediction horizon array. I added a clarification regarding that in the Methodology section.
>
> 3) While I am still working on additional dataset iterations (hopefully I can have full results soon), I added a traversal across different numbers of bins for the Binned Linear model for the ETTh1 dataset. With the periodic data, the number of bins equaling to periodicity aligns with the overall intuition of the choice.
>
> | **Bins** | **MSE (96)** | **MAE (96)** | **MSE (192)** | **MAE (192)** | **MSE (336)** | **MAE (336)** | **MSE (720)** | **MAE (720)** |
> | -------- | ------------ | ------------ | ------------- | ------------- | ------------- | ------------- | ------------- | ------------- |
> | 1 Bin    | 0.378        | 0.403        | 0.415         | 0.426         | 0.445         | 0.447         | 0.485         | 0.500         |
> | 2 Bins   | 0.378        | 0.396        | 0.412         | 0.419         | 0.443         | 0.442         | 0.479         | 0.495         |
> | 4 Bins   | 0.374        | 0.392        | 0.410         | 0.416         | 0.442         | 0.440         | 0.476         | 0.492         |
> | 6 Bins   | 0.373        | 0.391        | 0.410         | 0.415         | 0.442         | 0.438         | 0.472         | 0.489         |
> | 12 Bins  | 0.371 | 0.389 | 0.407  | 0.413  | 0.441  | 0.437  | 0.472  | 0.488  |
> | 24 Bins  | **0.369**    | **0.387**    | **0.405**     | **0.411**     | **0.437**     | **0.434**     | **0.470**     | **0.486**     |
> | 48 Bins  | 0.396        | 0.403        | 0.427         | 0.425         | 0.451         | 0.445         | 0.482         | 0.496         |
>
> As shown, performance improves monotonically as bins approach the true periodicity (24 hours for ETTh1), validating the theoretical framework. Performance degrades at 48 bins, demonstrating the trade-off between periodic alignment and statistical robustness.
>
> 4) I have added citations to the baseline models.
>
> 5) I hope to include additional results before the discussion deadline. However, I would like to note that the current baseline selection was intentional to provide interpretable and controlled validation of the theoretical framework. These models allow clear analysis of learned representations (through the weight representation, Section 5.1), which would be more obscured in complex architectures. The theoretical contribution (Theorems 1 and 2) and diagnostic framework (addition of the review, Section 5.3) are model-agnostic and provide the foundation for extending to complex architectures.

---

> > ### Comment · Reviewer_zRHK · 2025-11-26
> >
> > Although the perspective of modeling temporal periodicity via binning is interesting, the contributions and evaluation remain limited. I have adjusted my score accordingly.

---

### Official Review · Reviewer_snRQ · 2025-11-01

**Soundness:** 2
**Presentation:** 3
**Contribution:** 2
**Rating:** 4
**Confidence:** 3

**Summary:**

- The paper proposes a binning scheme that partitions the input window by phase with respect to an assumed period and trains per-bin linear mappings; horizon forecasts are formed by concatenating bin outputs.

**Strengths:**

- Clear seasonal/trend decomposition argument; bin-wise constant seasonal offset is intuitive
- The method is simple and portable, easy to reproduce and reason about.

**Weaknesses:**

- Even if bins match the true phase, the model forbids dependencies whose effective lags aren’t exact multiples of the period (e.g., holiday spillovers), so cross-phase signals alias or vanish. This induces approximation bias unless you add cross-bin coupling/regularization or a residual path that models inter-bin interactions.
- The narrative leans on rapid training-loss convergence on synthetic TO cases; this is not equivalent to systematic test improvement on real data (esp. in non-stationary time-series forecasting task)

**Questions:**

- Can you provide a diagnostic for when binning helps? E.g., estimate a trend-to-oscillation indicator on real data and correlate it with gains/losses (why ETTh2 drops)?
- What happens when the period is wrong, drifting, or multi-periodic? Can you show w!=b, learned/non-uniform bins, or online re-alignment?

---

> ### Author Response · Authors · 2025-11-22
>
> Thank you for the review and comments. Below are the answers to the questions and comments.
>
> 1) You are correct that the current formulation forbids effective lags that are not exact multiples of $w$. This is a fundamental trade-off: binning gains efficiency by assuming phase-independence but sacrifices the ability to model cross-bin interactions. There are potential extensions, which can incorporate binning with other techniques, for instance, a mixture architecture where binned predictions capture periodic structure and an unbinned component handles cross-phase interactions (as suggested in Section 5.5). For transformer-based extensions, there could be ideas of allowing limited cross-bin attention weighted by temporal proximity. The current results suggest this trade-off is favorable when periodic structure dominates (ETTh1, Electricity datasets) but detrimental when cross-phase dependencies are strong (ETTh2). I added the diagnostic metrics to quantify this in Section 5.3 (also defined in response to reviewer LqZz).
>
> | Dataset     | Period CV | Periodic Var% |
> |-------------|-----------|----------------|
> | ETTh1 | 0.050 | 56.41  |
> | ETTh2 | 0.109 | 14.27  |
> | Electricity | 0.010 | 82.54   |
>
> Periodic Var measures what fraction of total variance is explained by bin-specific means (i.e., periodic component that binning isolates), and Period CV measures consistency of periodic variation across time. ETTh2's weak periodic structure (14.27% vs 56.41% for ETTh1), combined with high CV (0.109), indicates the bins lack stable, distinct characteristics, potentially explaining consistent degradation.
>
> 2) The synthetic experiments demonstrate convergence efficiency and validate theoretical predictions about trend learning, but I agree that they do not prove generalization. The real dataset experiments are in Table 1. I have added the diagnostic metrics above specifically to bridge this gap, providing a principled way to predict real-data performance from data properties.
>
> 3) I have added bin ablation results showing performance as bins vary from 1 to 48 for ETTh1 (see response to Reviewer zRHK). Performance degrades gracefully when w is different than period, peaking at true periodicity (24). Regarding multiperiodic cases, I have extended Theorem 2 to show that binning can provide meaningful decomposition. Regarding drift, for non-stationary series where periodicity drifts, the current fixed-bin approach would degrade.

---

### Meta-Review · Area_Chair_pZad · 2025-12-17

**Summary:**

This research  proposes a  binned linear layer based time-series forecasting framework that based on that binning can serve as a simple means of efficient learning through isolating temporal patterns. The reviewers raise concerns regarding unclear technical details, over-strong assumption and limited experiments (implement experiments on  synthetic TO cases but rather than real-world series). This paper receives four negative evaluations and two of them have provided after-rebuttal feedback but still remain negative to this manuscript. Thus, based on comprehensive evaluations, especially the limitations mentioned above, this paper cannot be accepted at this current version.

**Reviewer Concerns:**

The reviewers raise concerns regarding unclear technical details, over-strong assumption and limited experiments (implement experiments on  synthetic TO cases but rather than real-world series).

**Reviewer Scores:**

The reviewers raise concerns regarding unclear technical details, over-strong assumption and limited experiments (implement experiments on  synthetic TO cases but rather than real-world series). This paper receives four negative evaluations and two of them have provided after-rebuttal feedback but still remain negative to this manuscript (e.g., improve 2 to 4, and remain 4)

---

### Decision · Program_Chairs · 2026-01-26

Reject